# Limited Lactosylation of Beta-Lactoglobulin from Cow’s Milk Exerts Strong Influence on Antigenicity and Degranulation of Mast Cells

**DOI:** 10.3390/nu13062041

**Published:** 2021-06-15

**Authors:** Gerlof P. Bosman, Sergio Oliveira, Peter J. Simons, Javier Sastre Torano, Govert W. Somsen, Leon M. J. Knippels, Rob Haselberg, Roland J. Pieters, Johan Garssen, Karen Knipping

**Affiliations:** 1Department of Chemical Biology and Drug Discovery, Utrecht Institute for Pharmaceutical Sciences, Utrecht University, Universiteitsweg 99, 3584 CG Utrecht, The Netherlands; G.P.Bosman@uu.nl (G.P.B.); J.Sastretorano@uu.nl (J.S.T.); R.J.Pieters@uu.nl (R.J.P.); 2Danone Nutricia Research, Uppsalalaan 12, 3584 CT Utrecht, The Netherlands; sergioooliveira@gmail.com (S.O.); Leon.Knippels@danone.com (L.M.J.K.); Johan.Garssen@danone.com (J.G.); 3Polpharma Biologics BV, Yalelaan 46, 3584 CM Utrecht, The Netherlands; Peter.Simons@PolpharmaBiologics.com; 4Division of Bioanalytical Chemistry, Amsterdam Institute for Molecules, Medicines and Life Sciences, Vrije Universiteit Amsterdam, de Boelelaan 1085, 1081 HV Amsterdam, The Netherlands; G.W.Somsen@vu.nl (G.W.S.); R.Haselberg@vu.nl (R.H.); 5Division of Pharmacology, Utrecht Institute for Pharmaceutical Sciences, Faculty of Science, Utrecht University Universiteitsweg 99, 3584 CG Utrecht, The Netherlands

**Keywords:** beta-lactoglobulin, Maillard reaction, lactosylation, cow’s milk allergy, antigenicity, mast cell degranulation

## Abstract

Background: beta-lactoglobulin (BLG) is one of the major cow’s milk proteins and the most abundant allergen in whey. Heating is a common technologic treatment applied during milk transformational processes. Maillardation of BLG in the presence of reducing sugars and elevated temperatures may influence its antigenicity and allergenicity. Primary objective: to analyze and identify lactosylation sites by capillary electrophoresis mass spectrometry (CE-MS). Secondary objective: to assess the effect of lactosylated BLG on antigenicity and degranulation of mast cells. Methods: BLG was lactosylated at pH 7, a water activity (aw) of 0.43, and a temperature of 65 °C using a molar ratio BLG:lactose of 1:1 by incubating for 0, 3, 8, 16 or 24 h. For the determination of the effect on antibody-binding capacity of lactosylated BLG, an ELISA was performed. For the assessment of degranulation of the cell-line RBL-hεIa-2B12 transfected with the human α-chain, Fcε receptor type 1 (FcεRI) was used. Results: BLG showed saturated lactosylation between 8 and 16 incubation hours in our experimental setup. Initial stage lactosylation sites L1 (N-terminus)—K47, K60, K75, K77, K91, K138 and K141—have been identified using CE-MS. Lactosylated BLG showed a significant reduction of both the IgG binding (*p* = 0.0001) as well as degranulation of anti-BLG IgE-sensitized RBL-hεIa-2B12 cells (*p* < 0.0001). Conclusions and clinical relevance: this study shows that lactosylation of BLG decreases both the antigenicity and degranulation of mast cells and can therefore be a promising approach for reducing allergenicity of cow’s milk allergens provided that the process is well-controlled.

## 1. Introduction

Food allergens have the potential to induce various health concerns in susceptible individuals. The majority of allergenic foods are usually subjected to thermal processing prior to their consumption since it is a cost-efficient treatment widely used by industries to achieve food safety and extended shelf life [1]. During this thermal processing and subsequent long storage of foods, the Maillard reaction (MR) frequently occurs. The MR is a non-enzymatic glycation reaction between the carbonyl compound of reducing sugars and compounds having amino groups. The MR may sometimes be beneficial by damaging epitopes of allergens and reducing their allergenic potential [2,3,4,5,6,7], while the opposite, i.e., exacerbation of allergic reactions, as a result of neo-allergen generation may occur as well [7,8]. Apart from these modulations, non-enzymatic glycation can modify food proteins with various types of advanced glycation end products (AGEs) such as Nϵ-(carboxymethyl-)lysine (CML), pentosidine, pyrraline, and methylglyoxal-H1 derived from the MR [9]. These MR products may act as immunogen by inducing the activation and proliferation of various immune cells [3,10,11]. Ample evidence is available about the pathogenesis caused by glycation in the context of various diseases, but there are a few studies that provide a thorough insight into the impact of lactosylation on food allergies [3,8,12].

Alteration of the allergenicity of cow’s milk proteins by glycation has previously been reported [13]. Alpha-lactalbumin (ALA) was maltopentaosylated in a MR type setup and subsequently phosphorylated by dry heating in the presence of pyrophosphate, to investigate its structure and physiological functions. The relative sugar content of ALA was increased to approximately 22.3%. The affinity of an anti-ALA antibody was significantly impaired. The suppressive effect of ALA on the production of proinflammatory cytokines such as interleukin-6 and tumor necrosis factor-α from THP-1 cells after stimulation with lipopolysaccharide was significantly enhanced by glycation with maltopentaose and was further enhanced by phosphorylation after glycation [13]. However, when beta-casein, another milk protein, reacted with D-glucose using reductive amination, both the structural and functional aspects of beta-casein were altered, but the allergenicity profile of this protein remained largely unchanged [14].

Beta-lactoglobulin (BLG) is one of the major cow’s milk proteins and the most abundant protein in the whey fraction. In its native state, this 18 kDa globular protein forms a beta-barrel (or calyx) structure. The structure is stabilized by two intra-molecular disulfide bonds and can be altered by heating above 65 °C [2]. Although heat is frequently applied during milk processing, the effects of heat-induced denaturation of BLG on its recognition by immunoglobulin E (IgE) from cow’s milk allergy (CMA) patients are not fully understood. In previous studies, heat-induced denaturation of BLG was associated with weaker binding to IgE from CMA patients. Heat treatment reduced the allergenicity of BLG by inducing conformational changes and by increasing its susceptibility to enzymatic digestion, which resulted in disruption of B-cell epitopes [15]. Moderate modification of BLG in the early stages of the MR only had a small effect on its recognition by IgE, whereas a high degree of glycation has a clear “masking” effect on the recognition of epitopes. This demonstrates the importance of ε-amino groups of lysines, the site of glycation, in epitopes that are recognized by IgE [2]. In addition, the effects of dynamic high-pressure microfluidization (DHPM) treatment and Maillardation in the presence of galactose on the IgE-binding capacity and conformation of BLG were investigated [8,16]. The binding capacity of IgE from CMA patients’ sera with galactosylated BLG decreased after DHPM treatment.

Glycation and heat treatment of other food allergens such as buckwheat [4], chickpea [12], hen’s egg ovalbumin and ovomucoid [6,17,18], peanut [7], hazelnut [11] and shrimp [19] have also been studied and showed an impact on immunoreactivity and allergenicity. It was demonstrated that immunoreactivity to glycated proteins is highly variable; it may decrease, remain unchanged, or even increase after food glycation.

Although the MR is generally perceived as undesirable and should therefore be controlled during food processing, glycation may be a promising approach for reducing the allergenicity of food allergens as glycation might offer the opportunity to manipulate the allergenic properties of proteins. In a detailed physicochemical and functional study, the effect of glycation on the antigenicity and allergenicity of BLG from cow’s milk was assessed. In the current study, lactose was chosen for the glycation of BLG as it is the most abundant milk sugar. A complete overview of the MR of BLG was obtained by performing capillary electrophoresis mass spectrometry on both intact and trypsin-digested BLG to study preferential sites for lactosylation. This information was linked to the epitopes that are recognized by the antibodies used in antigenicity and degranulation assays.

## 2. Materials and Methods

### 2.1. Materials

Bovine BLG A (BLGA), bovine BLG B (BLGB) and wildtype BLG (BLGWT) consisting of both BLGA and BLGB were obtained from Sigma-Aldrich, St. Louis, MO, USA (see Appendix A for sequences of each of the isoforms). For SDS-PAGE, a Bio-Rad Mini-PROTEAN II system was used with a 4 to 20% TGX gel (Bio-Rad Laboratories, Hercules, Clearwater, FL, USA). Sample buffer consisted of 0.2 M Tris-HCl (Merck, Darmstadt, Germany), 277 mM SDS (Boehringer Mannheim, Mannheim, Germany), 0.4 M dithiothreitol (ICN), 6 mM Bromophenol blue (Merck, Darmstadt, Germany), 4.3 M glycerol (Merck, Darmstadt, Germany), pH 6.8. Trans-blot turbo mini PVDF transfer pack (Bio-Rad Laboratories, Hercules, Clearwater, FL, USA) was used with the Bio-Rad Transblot Turbo System. For Western blotting, gelatin (Merck, Darmstadt, Germany), anti-BLG polyclonal antibodies (A10-125P; Bethyl Laboratories, Montgomery, AL, USA) and Lumi-light Plus Western blotting substrate (Boehringer Mannheim, Mannheim, Germany) were used. The molecular weight marker was the pre-stained Precision Plus Protein Standards Dual Xtra, 161-0377 (BioRad Laboratories, Hercules, Clearwater, FL, USA). TBS-T consisted of 20 mM Tris (Sigma-Aldrich, St. Louis, MO, USA) and 150 mM NaCl (Merck, Darmstadt, Germany) and 0.1% Tween 20 (Sigma-Aldrich, St. Louis, MO, USA), pH 7.6. The capillary electrophoresis bare-fused silica capillary (inner diameter of 50 μm and an outer diameter of 365 μm) from Biotaq (Silver Springs, MD, USA) was coated with polybrene (PB) and dextran sulfate (M_r_ ~40,000 Da; DS) from Sigma-Aldrich, St. Louis, MO, USA. Other chemicals were domestic products of analytical grade. For the RBL degranulation-assay, human purified IgE (AG30P, Merck, Darmstadt, Germany), rabbit anti-human IgE antibodies (A009402, Agilent, Santa Clara, CA, USA), a pool of BLG-specific chimeric IgE monoclonal antibodies (Polpharma Biologics, Utrecht, The Netherlands), human serum albumin (A1887, Sigma-Aldrich, St. Louis, MO, USA) and β-hexosaminidase substrate 4-methylumbelliferyl-N-acetyl-α-D-glucosamine (4-MUG; Sigma-Aldrich, St. Louis, MO, USA) were used. Tyrode’s buffer consisted of 130 mM NaCl (Merck, Darmstadt, Germany), 5 mM KCl (Merck, Darmstadt, Germany), 1.4 mM CaCl_2_ (Sigma-Aldrich, St. Louis, MO, USA), 1 mM MgCl_2_ (Sigma-Aldrich, St. Louis, MO, USA), 5.6 mM glucose (Merck, Darmstadt, Germany) and 10 mM N-2-hydroxyethylpiperazine-N-2-ethanesulfonic acid (HEPES; Merck, Darmstadt, Germany), pH 7.4.

### 2.2. Sample Preparation

The lactosylated samples were obtained accordingly to a methodology previously described [20]. Using a mass ratio of 1:1, BLGWT and lactose were dissolved in deionized water at approximately 30% (*w*/*w*); the pH was adjusted to pH 7 or 9 with 1N NaOH. After freeze-drying, samples were immediately placed in evacuated desiccators in the presence of anhydrous P_2_O_5_ for two weeks to ensure a water activity (a_w_) of 0.43. Lactosylation was conducted in a temperature-controlled oven at 65 °C for several time periods (3, 8, 12 and 24 h). The control sample t = 0 was not heated, but underwent the same processing up until the analyses. Subsequently, for each interval, the BLGWT/lactose mixture was dissolved in water and buffer exchanged to pure MilliQ water using 3 kDa MWCO spin filters (Amicon/Millipore) and was concentrated to 2 mg/mL. Reference proteins BLGA and BLGB were not modified and were directly dissolved in MilliQ water at 2 mg/mL and stored at −20 °C until further use. In general, for glycopeptide analysis, 200 μg protein was denatured using 8 M urea, reduced in 15 mM DTT, and subsequently alkylated with 45 mM iodoacetamide; each step was incubated for 1 h at room temperature. After a 10-fold dilution, samples were digested by trypsin (Promega—V5111) (50:1, protein:trypsin) in ammonium bicarbonate 100 mM pH 8 during overnight incubation at 37 °C. A Sep-Pak C18 1CC Vac SPE Cartridge (Waters) was used to desalt the sample, and the eluate was lyophilized followed by reconstitution in 50 μL water.

### 2.3. Sodium Dodecylsulfate-Polyacrylamide Gel Electrophoresis (SDS-PAGE) and Western Blotting (WB)

For SDS-PAGE, proteins were diluted in a 1:3 ratio with reducing Laemmli sample buffer (Sigma-Aldrich) and 1 µg/lane of protein was loaded on the gel. After protein separation, the proteins were transferred to a PVDF membrane. PVDF membranes were blocked with 2% gelatin in TBS-T and incubated with an HRP-conjugated rabbit anti-bovine BLG antibody 1:30,000 in 0.5% gelatin in TBS-T. Antibody binding was visualized by using Lumi-light Plus Western blotting substrate and chemiluminescence signals were measured with the Chemidoc XRS (Bio-Rad).

### 2.4. Capillary Electrophoresis—Mass Spectrometry

A Beckman Coulter Proteomelab PA-800 capillary electrophoresis (CE) instrument, coupled with electrospray ionization (ESI) to a Bruker Daltonics QII quadrupole-time-of-flight (QTOF) mass spectrometry (MS) detector, was used for analysis. The CE-MS interface was an Agilent Technologies co-axial sheath-liquid (SL) sprayer interface with an SL consisting of 50% methanol, 49.9% water and 0.1% formic acid (*v*/*v*/*v*), delivered by a KD Scientific Syringe Pump using a gas-tight glass syringe of 1 mL with a flow rate of 300 μL/h. Acquisition parameters were set as follows: nebulizer gas (nitrogen) 5 psi, drying gas 4 L/min, drying temperature 180 °C, capillary voltage 4500 V, end plate offset −500 V, dry heater scan range in positive mode 500 to 4000 *m*/*z*. Glycopeptide analysis was carried out by coupling the same CE-ESI setup to an Agilent 1100 Series ion trap (IT) MS. Interface settings for dry gas temperature were 300 °C and for nebulizer gas (nitrogen) 5 L/min. Transfer parameters were optimized for the mass range of 100 to 2200 *m*/*z* using an ESI tuning mix in case of the IT.

### 2.5. Intact Protein Analysis

Mass spectrometry spectra of intact proteins were obtained through measurements on the QTOF system and spectra were deconvoluted using Bruker Daltonics Maximum Entropy deconvolution software. Settings of a mass range of 10 to 20 kDa, data point spacing of 0.1 *m*/*z* and maximum resolving power at high resolution, at which mass differences of 324.10 Da in the deconvoluted spectra could be observed (corresponding to the attachment of a lactose moiety minus a released water molecule of 18.01 Da), were used. Assuming all pairs of cysteine residues form cystines in native condition, the theoretical masses for BLGA and BLGB ought to be 18,363.16 Da and 18,277.07 Da, respectively.

### 2.6. Peptide Identification

The output of the MS analysis of the digests is expected to be elaborate; therefore, a theoretical library was created, containing the theoretical expected peptides formed during trypsin hydrolysis. Peptides from both BLGA and BLGB with a charge from 1+ to 6+ are incorporated, as are miscleavages and incompletely reduced and alkylated digests and potential modifications, such as lactosylation. In total, the library consisted of approximately 6000 potential peptides and was used to identify peptide peaks, generated with CE-ESI-IT, by *m*/*z* value, within ±0.2 *m*/*z*. Comparison of the relative intensities of the peaks representing peptides with attached lactose units was performed to gain insight of the favorable lactose attachment sites and the development of the lactosylation profile during the five time intervals.

### 2.7. Antigenicity Assessment of BLG

For the determination of the effect of antibody-binding capacity (antigenicity) of the lactosylated BLG samples, the Beta-Lactoglobulin ELISA Kit (ELISA Systems; ESBLG-96) was performed according to the manufacturer’s protocol. This sandwich-type ELISA has been especially developed for the detection of BLG residues in food products and has a range of 0.1 to 1.0 part per million (ppm). The antibody used is a polyclonal antibody raised against BLG purified from bovine milk (>90% pure by SDS-PAGE). Lactosylated BLGWT was weighed, dissolved, and diluted to a final concentration of 80 μg/mL in extraction buffer, which is supplied in the kit.

### 2.8. Degranulation of Rat Basophilic Leukemia-huFcεRI Cells (RBL-hεIa-2B12 Cells)

The cell-line RBL-hεIa-2B12, which was transfected with the α-chain of the human Fcε receptor type 1 (FcεRI), was used for the RBL-huFcεRI assay to assess the degranulation capacity of the lactosylated BLG samples. Confluent growing RBL-huFcεRI cells (1 × 10^5^/well) in a 96-well flat-bottom culture plate were sensitized overnight with 5 μg/mL commercially available purified human IgE and stimulated with 10 μg/mL cross-linking rabbit anti-human IgE antibodies in Tyrode’s buffer at pH 7.4 with 0.1% HSA for 1 h. This release served as the maximum release (100% degranulation). Anti-BLG specific chimeric mouse/human IgE sensitized RBL-huFcεRI cells [21,22] were stimulated with cross-linking rabbit anti-human IgE (10 µg/mL) or lactosylated BLG samples 1 μg/mL in Tyrode’s buffer/HSA for 1 h. β-Hexosaminidase activity was determined by a fluorescence assay using 4-MUG as a substrate. The β-hexosaminidase released into the medium (degranulation marker) was expressed as the percentage of maximum release observed after cross-linking with anti-human IgE antibodies.

### 2.9. Statistics

Comparisons of IgG binding (BLG ELISA) and degranulation (RBL-hεIa-2B12 cells assay) were performed with one-way ANOVA, which is used to determine statistically significant differences between the means of two or more independent groups using GraphPad Prism 8.0.0. *p*-values < 0.05 were considered statistically significant.

## 3. Results 

### 3.1. Lactosylation of BLG and Characterization by WB

After 24-h freeze-drying of the BLG/lactose mixture, large particles were ground and placed in a desiccator for two weeks. The powder mixture was placed in a prewarmed oven to obtain a total of 10 samples—five samples for each pH condition. The Western blot after an SDS-PAGE run under reducing conditions of BLG and its conjugates are shown in Figure 1. Compared with the unheated BLG (t = 0), the molecular weight of lactosylated BLG time-dependently increased with heating and stabilized after t = 16, suggesting that a coupling reaction occurred between BLG and lactose and reached a maximum. Several aggregation bands corresponding to the oligomeric forms were observed, indicating that covalent cross-linkage reactions at all time points occurred. No discernable difference between pH 7 and pH 9 on gel could be distinguished; hence, samples acquired by lactosylation at pH 7 were used for further analyses.

### 3.2. CE-MS Analyses of Intact BLGWT, BLGA and BLGB

Prior to each run, the 90-cm coated capillary was rinsed for 5 min (min) at 20 psi with background electrolyte (BGE; see Appendix A for the coating procedure). After that, the sample was injected using 1 psi for 10 s, resulting in a volume of analyte of 0.86% (<1%) of the total capillary volume corresponding to 150 nL. The runtime varied between 30 min and 1 h. Due to the application of the coating, bearing a positive charge, the electro-osmotic flow (EOF) changed direction. Hence, a reversed polarity was applied to ensure an EOF towards the mass spectrometer. Choosing a low pH BGE ensures that the protein is also positively charged and the analyte–wall interaction is prevented. Under these conditions, a sufficient separation of the two protein isoforms (R = 0.88, r = resolution; for more details see Appendix A) was obtained. The isoforms only differ in two amino acids; this mass difference can be easily discerned in the deconvoluted mass spectra obtained in the separated peaks (Figure 2). BLGA and BLGB standards were used to confirm the migration times of the peaks in the BLGWT sample.

After charge deconvolution, masses of 18,363.4 Da and 18,277.0 Da were found for BLGA and BLGB, respectively, indicating a charge distribution of 9+ to 15+, as shown in the profile spectrum (middle) of Appendix A. The observed masses are in line with the calculated mass values of BLGA and BLGB, as shown in Appendix A. Next, the lactosylated samples have been characterized by intact protein MS, of which the deconvoluted results are shown in Figure 3. When the lactosylation levels increased, the net charge decreased, caused by linkage of lactose amine groups, shielding positive charges, resulting in shorter migration times and lower MS responses as incubation times increased. Apparently, at t = 0, some lactosylation already occurred spontaneously. At t = 3 and t = 8, a shift to the higher masses can be observed, no non-lactosylated protein remained and all protein contained 3, 4, 5, 6, 7, 8 or 9 lactose units. The mass shift continued until t = 24, with overlapping lactosylation levels, where the difference in lactose units per protein was only 1 lactose unit between t = 16 and t = 24; seemingly, a maximum is reached.

The protein BLG has 19 potential lactosylation sites, namely the N-terminal leucine, 15 lysine and three arginine residues. A range of lactose units per molecule was observed between incubation times of 3 to 16 h, while heterogenicity lessened after 16 h. A plateau was reached after 8 to 16 h incubation, when 16 sites were lactosylated. Figure 4 shows the distribution of lactose units per BLG molecule over time. 

### 3.3. Analyses of Tryptic Digest of BLG

To investigate the initial binding preferences of lactose, we focused on the first three time intervals: t = 0, t = 3 and t = 8. Reduced, alkylated and trypsin-digested BLGWT and conjugates (lactosylated at pH 7) were analyzed by CE-ESI-IT and compared to the generated peptide database. An amino acid sequence recovery of 100% was obtained for all time points. Glycopeptides containing one or two trypsin miscleavages (due to lactose blockade) and ≥1 lactose units per peptide were selected (Appendix A) and relative intensities of reoccurring glycopeptides at the three intervals revealed the preferential sites for lactosylation (Figure 5). When corresponding peptides of BLGA/B appeared, an average intensity was used for calculations of percental increase over time. In line with the intact BLG analyses, some peptides at t = 0 were identified as lactosylated, though at very low levels. These lactosylated peptides at t = 0 were set as baseline. The relative increase of lactosylation during each interval was based on the level of lactosylation at the starting point of that interval. In that way, as shown in Figure 5 (and Appendix A), the apparent most accessible lactosylation sites could be identified after 3 h incubation if compared with the non-incubated BLGWT (t = 0) (not shown). After 8 h incubation, with respect to t = 3, the already existing lactosylated peptides were observed more abundantly, and some more lactosylation sites could be identified.

### 3.4. Antigenicity of Lactosylated BLG Using ELISA

To evaluate the effect of lactosylation of BLG on IgG binding and thereby assessing antigenicity, a BLG ELISA was used. Lactosylation showed a significant (ANOVA *p* = 0.0001) lactosylation-dependent reduction of antigenicity (Figure 6). At t = 3, antigenicity strongly decreased to 50%, and decreased even more, with about 80% at t = 8. The time points t = 16 and t = 24 showed similar binding to IgG with a reduction of antigenicity of about 90%, demonstrating that lactosylation of BLG can interfere with the binding to IgG antibodies and thereby reduce the BLG antigenicity.

### 3.5. Degranulation Capacity of Lactosylated BLG Using RBL-hεIa-2B12 Cells

To evaluate the effect of lactosylation of BLG on IgE-cross-linking and thereby assessing allergenicity, the degranulation of anti-BLG IgE-sensitized RBL-hεIa-2B12 cells was measured (Figure 7). Since no lactosylation differences were found after 16 h in the ELISA and WB, sample t = 24 was excluded from allergenicity testing. A pool of IgE antibodies was used that recognizes most of the sequence of BLG, except L1–T18 and P79-I84, shown grey in the sequence displayed in Figure 5. In this way, the effect of lactosylation near the N-terminus was not assessed. However, lactosylation of BLG showed a significant (ANOVA *p* < 0.0001) reduction of degranulation by RBL-hεIa-2B12 cells, which was reduced by 30% with BLG that was lactosylated for 3 h, 40% with lactosylated BLG at t = 8 and 55% with lactosylated BLG at t = 16, demonstrating that lactosylation of BLG can reduce its allergenicity.

## 4. Discussion

During the last few decades, the incidence of food allergies has increased in developed countries, and it is suggested that the Western lifestyle and diet promote innate danger signals and immune responses through the production of “alarmins”. Alarmins are endogenous molecules secreted from cells undergoing nonprogrammed cell death that signal tissue and cell damage. Advanced glycation end products present in food might predispose humans to food allergies [22]. The majority of allergenic foods are usually subjected to thermal processing prior to their consumption. However, during thermal processing and the long-term storage of foods, MR products are frequently generated. Recent studies have suggested that the MR products, in particular AGEs, present in human diets may be involved in the development of chronic inflammation by acting as inflammatory components and affecting the gut microbiome [5]. However, MRs may sometimes be beneficial by damaging the epitopes of allergens and thereby reducing their allergenic potential, while the exacerbation in allergic reactions may occur as well due to changes in the motifs of epitopes or neo-allergen generation [9]. 

Peptide-driven profiling (data not shown) of the used pool of BLG-specific chimeric IgE monoclonal antibodies has revealed binding to overlapping regions covering residues W19-I78 and D85-I162 of BLG. Structural analyses of Protein Data Bank files 2Q2M [23], 1CJ5 [24] and 1BEB [25] indicate that nearly all lysines and arginines are solvent-exposed and are expressed on the outer surface of BLG. Based on these structures, no preferential lactosylation sites can be identified. However, after trypsinization and semi-quantitative analyses, it was apparent that some sites of BLG are more prone to lactosylation than others. As already known from the intact protein analyses, some spontaneous attachment of lactose occurred at t = 0. The 3-h incubation clearly showed preferences in lactosylation sites. More specifically, K141, K138, K77 (in order of abundancy) and the N-terminal were lactosylated in the first 3 h. All these lysines, except the N-terminus, are present in the epitopes recognized by the chimeric IgE ɑBLG antibodies, and are lactosylated, which may explain the reduced allergenicity in the RBL assay. After 3 h, less elevated lactosylation sites K91, K75, K60 and K47 were identified in the peptide analysis. Adding another 5 h of incubation to the sample, further intensification of the signal of the two aforementioned sites such as the N-terminus (tryptic peptide L1-K8 is now also observed with one and two lactose moieties; hence, K8 must have become lactosylated) and K47 was seen. Most likely, K60 also becomes lactosylated as the corresponding tryptic peptide V41–K60 is found with one and two lactose moieties. K101, K69, and R148 mostly become lactosylated after 8-h incubation. Approaching the lactosylation plateau at 8 h, with 8 to 15 lactose units per molecule BLG, most other regions that were not yet lactosylated at t = 3 of BLG have been lactosylated. However, the area corresponding to G9-R40 seems to remain non-lactosylated under our examined conditions.

It has been widely reported that the lactosylation of proteins can reduce the allergenicity of these proteins [2,3,4,6,7,11,12,14,16,17,26,27] but there is little knowledge on which specific lactosylation sites might have an effect on the allergenicity of proteins. A recent study describes that when tropomyosin (TM) from shrimp was glucosylated (TM-G), this led to weaker allergic responses in mice and mast cells, which might be explained by glucose destroying the epitopes with glucosylation sites. The glucose residues on TM-G functioned as both a resistance to prevent the accessibility of TM-G to the IgE on mast cell and masking epitopes [19]. In our study, we found a decrease in both the antigenicity and allergenicity of lactosylated BLG. For the antigenicity assessment using the BLG ELISA, we found that lactosylation with a temperature of 65 °C showed a significant reduction of IgG antibody-binding capacity with increased lactosylation, with an optimal reduction with aw 0.43 and no difference found between pH 7 or pH 9, and no additional reduction after 16 h. Over the whole course of the preparation of the conjugates, dimers, and trimers of BLG were formed and observed on WB. Protein aggregation might have an effect on solubility as well as on antibody binding. In the case of BLG, solubility was not an issue at all time intervals; all samples were well-soluble. However, reduced epitope availability as a result of aggregation was taken into account. If aggregation—as a result of incubation at elevated temperatures for extended time periods—would affect outcomes in ELISA, a similar inhibition of signals is expected as equal aggregation was observed at each of the intervals on WB. In many studies, binding of allergens to IgE in serum from allergic patients is used to assess allergenicity. In our study, we used a cell-based assay that can study the cross-linking of IgE by allergens and concomitant degranulation of the cells, which is a more physiological assessment of allergenicity than binding to IgE alone. It was found that the condition with the best potential for reducing the degranulation of RBL-hεIa-2B12 cells were of lactosylated BLGWT incubated at 16 h under given conditions of lactosylation. Under reducing conditions, the Western blot profiles of lactosylated BLG shows a clear increase in weight of the monomer, dimer and trimer forms compared with the unheated BLG (t = 0). All masked sites except the N-terminus and K8 on the first two intervals (t = 0 to t = 8) were within the area that is covered by the pool of BLG-specific chimeric IgE monoclonal antibodies, explaining the continuation of the reduction of degranulation in subsequent experiments, evidently showing that a reduction in recognition by antibodies and subsequent reduced antigenicity/degranulation is caused by increased lactosylation.

## 5. Conclusions

In conclusion, the current study demonstrated that lactosylation at 65 °C, aw 0.43, pH 7 for 16 h decreased both the antigenicity and degranulation capacity of BLG and can therefore be a promising approach for reducing the allergenicity of cow’s milk allergens, provided that the process is well-controlled.

## Figures and Tables

**Figure 1 nutrients-13-02041-f001:**
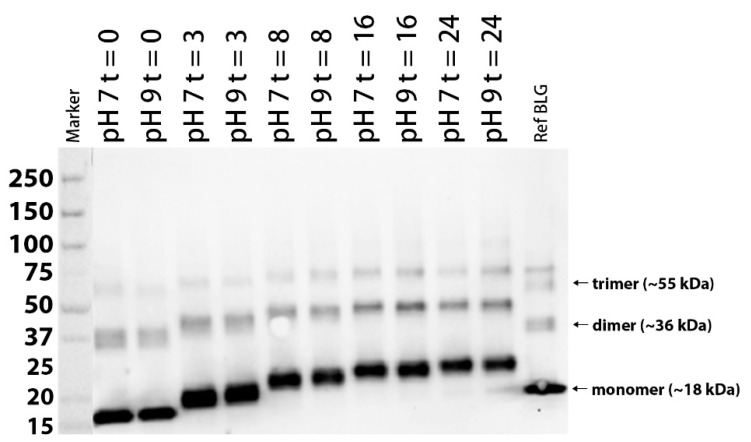
Western blot of reference BLG (Davisco JE 003-6-922) and lactosylated BLGWT (lacBLG) after 0, 3, 8, 16 and 24 h incubation at pH 7 and 9 and 65 °C. Protein ladder is expressed in kDa. Equal formation of dimers and trimers of BLG was evident during the time course. Comparable to monomers, lactosylation can be observed amongst dimers and trimers.

**Figure 2 nutrients-13-02041-f002:**
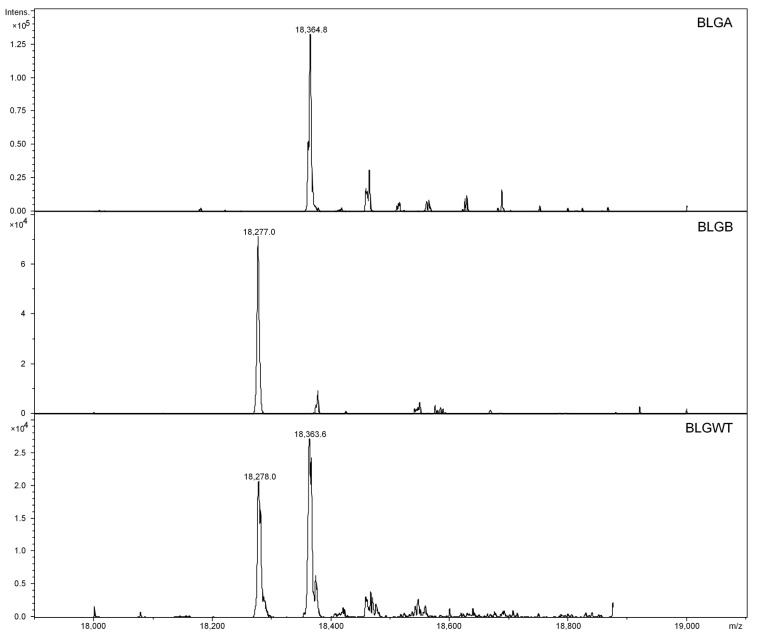
Deconvoluted spectra of BLGA, BLGB and BLGWT.

**Figure 3 nutrients-13-02041-f003:**
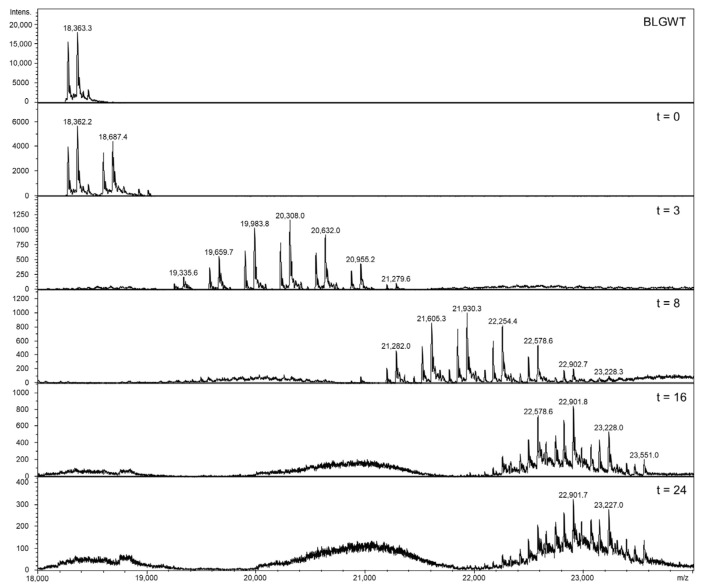
Deconvoluted spectra of BLGWT and the conjugates at different time intervals. A clear shift of 324.1 Da is visible at each time interval, which can be attributed to an increase of one lactose residue (342.1 Da) and the loss of water (18.01 Da) during Maillardation. Signal intensities decrease with increasing lactosylation levels. See Appendix A for accurate masses.

**Figure 4 nutrients-13-02041-f004:**
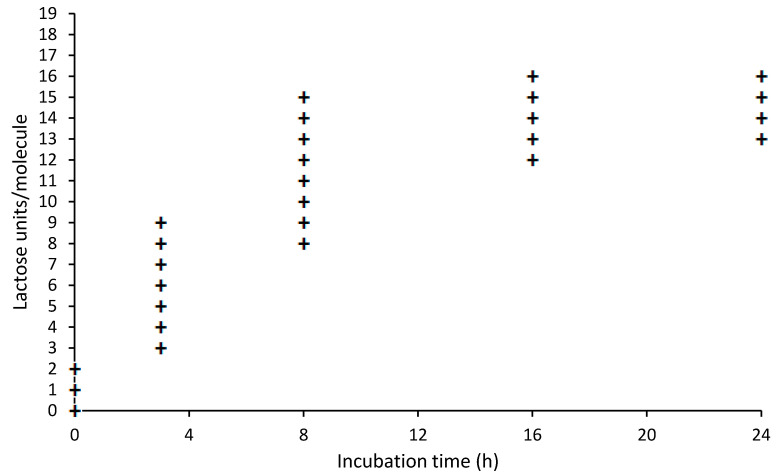
Lactose units (+) per BLG molecule. BLG seems to reach a maximum of 16 lactose moieties per molecule after 16 h. In the initial phase, 0 to 8 h incubation, varying amounts of lactose per protein can be observed.

**Figure 5 nutrients-13-02041-f005:**
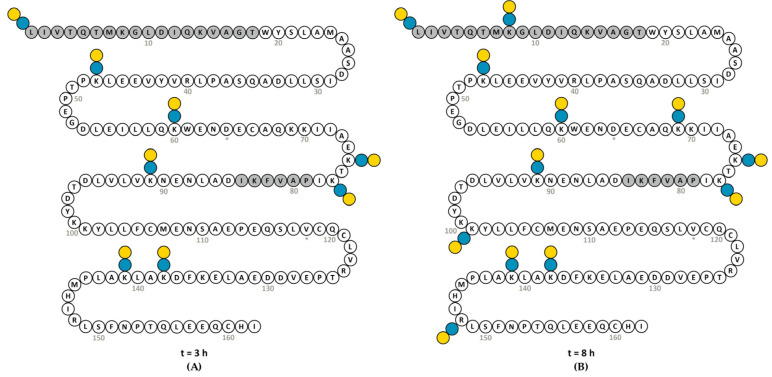
(**A**): after 3 h, lactosylation occurred at L1 (N-terminus), K47, K60, K75, K77, K91, K138 and K141 of BLGWT. (**B**): after 8 h additional sites K8, K69, K101 and R148 were lactosylated. Lactose moieties are indicated by yellow/blue bispheres. In grey: non-binding areas of the pool of BLG-specific chimeric IgE monoclonal antibodies used in the RBL assay. * BLGB = BLGA D64G, V118A.

**Figure 6 nutrients-13-02041-f006:**
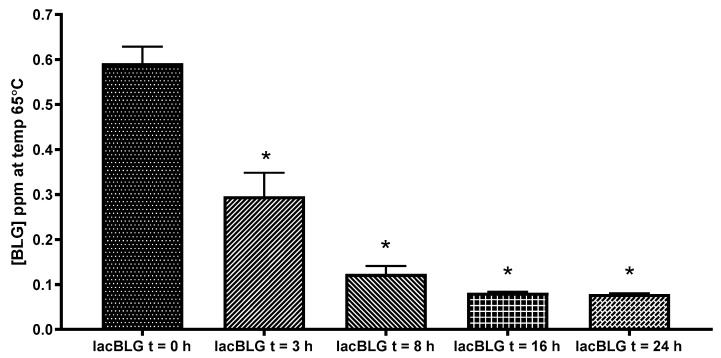
IgG binding (antigenicity) of lactosylated BLGWT (lacBLG) in duplicate using a BLG ELISA. Data are expressed as mean + SD.* One-way ANOVA *p* = < 0.0001.

**Figure 7 nutrients-13-02041-f007:**
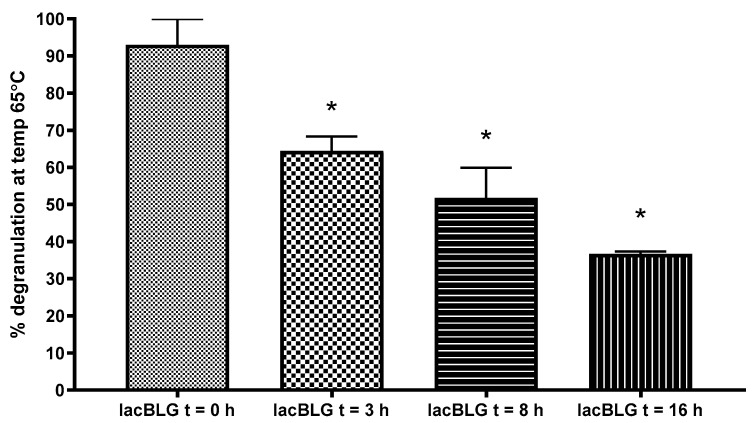
Human FceRI a chain expressing RBL cells incubated with a pool of chimeric huIgE anti-BLG antibodies, and subsequently cross-linked with anti-hIgE antibodies, which is set as maximum (100%) degranulation (bar not shown) or lactosylated BLGWT (lacBLG) in triplicate. Degranulation (surrogate marker of IgE binding ability or allergenicity), measured by release of β-hexosaminidase, of lactosylated BLG was calculated in % of the maximum degranulation. * One-way ANOVA *p* = < 0.0001.

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
