# Peer review of "Limited Lactosylation of Beta-Lactoglobulin from Cow’s Milk Exerts Strong Influence on Antigenicity and Degranulation of Mast Cells"

_nutrients, 2021, doi:10.3390/nu13062041_

Round 1

Reviewer 1 Report

The revised manuscript and author's point-by-point response are satisfactory to this reviewer.

Author Response

Thank you to the reviewer for his/hers valuable comments and agreeing with the manuscript in its current form

Reviewer 2 Report

There are still major concerns to address

The results show a reduced IgE-binding capacity. Allergenicity should be determined by the use of serum from allergic and non-allergic patients (see also line 316).

  1. The binding ability of IgE is not the same as allergenicity due to the fact that IgE bound to the high affinity FceRI not necessarily leads to a degranulation of e.g. RBL cells.
  2. According to the European Food Safety Authority (efsa) allergenicity is “the ability to trigger an abnormal immune response that leads to an allergic reaction in a person.” That means that an immune response is triggered. The RBL assay solely shows a degranulation upon FceRI-cross-linking by the use of (artificial) IgE antibodies indicating their ability to receptor- and antigen-binding (meaning functionality).
  3. As the FceRI-receptor represents the high affinity receptor for IgE it is most unlikely that IgE “are not able to bind”. Additionally, the here used “RBL-hεIa-2B12 cells” have been studied by Ladics et al. (cited in the mentioned Knipping K. et al. Intra- and inter-laboratory validation of an innovative huFcepsilonRIalpha-RBL-2H3 degranulation assay for in vitro allergenicity assessment of whey hydrolysates. Toxicol in vitro 2016) where it has been found that 80-100% of the studied RBL cell lines bound human IgE.
  4. Pool sera from CMA patients are not necessarily mandatory but sera from CMA patients that are well diagnosed (by e.g. component resolved diagnostics) are available.
  5. The authors doubt the sensitivity of the here described RBL assay using sera with “1 specific allergen” but being “very specific and sensitive assay” using the chimeric antibodies. What exactly is the outcome of this assay proving “allergenicity” with chimeric (assuming mouse variable domains fused to human IgE-Fc) antibodies instead fully human IgE from patients?
  6. Why then are the developed “chimeric IgE antibodies for the use in the RBL assay for the safety assessment of whey hydrolates for CMA patients” when no data with IgE from CMA patients exist (not even in western blot analyses)?
  7. Moreover, as stated in lines 352 - 354 (“It is widely reported that lactosylation of proteins can reduce the allergenicity of these proteins [2-4, 6, 7, 11, 12, 14, 16, 17, 26, 27] but there is little knowledge on how lactosylation might have an effect on the allergenicity of proteins.” The question of “how”, actually, has not been answered.

Minor concerns

Line 185: In a sandwich ELISA “Detection” is usually performed by the use of a secondary antibody conjugated to a e.g. chromophore.

Line 248: Please change “non-lactolated” to “non-lactosylated”.

Line 276: Please change “percentual” to “percental”

Line 278: t = 0 is still not shown (“As shown in figure 5”).

Lines 276 - 278: Please explain why “the apparent most accessible lactosylation sites could be identified after 3 h incubation if compared with the non-incubated BLGWT (t = 0)”.

Line 298: Is it “+SD”? Please check.

3.5 has not been revised: Assuming that the different points in time (e.g. t=0) shown in figure 7 indicate the incubation time of the BLGs with lactose it is not a “time-dependent reduction of degranulation” but rather a lactosylation-dependent reduction. Please revise point 3.5.

Figures 6 and 7: Please specify the asterisks.

Line 356: Please erase “Moreover”.

Lines 359/360: Does lactosylation show good solubility? Please check and revise the sentence.

Lines 362 - 365: Please check syntax.

Lines 369 - 371: Please check syntax.

Line 370: Please specify the samples that are meant.

Line 374: “αBLG IgE pool” is mentioned for the first time. Please clarify this “αBLG IgE pool” or paste this expression to line 115 after “pool of BLG-specific chimeric IgE monoclonal antibodies (Polpharma Biologics)”

Author Response

There are still major concerns to address

The results show a reduced IgE-binding capacity. Allergenicity should be determined by the use of serum from allergic and non-allergic patients (see also line 316). The binding ability of IgE is not the same as allergenicity due to the fact that IgE bound to the high affinity FceRI not necessarily leads to a degranulation of e.g. RBL cells. According to the European Food Safety Authority (efsa) allergenicity is “the ability to trigger an abnormal immune response that leads to an allergic reaction in a person.” That means that an immune response is triggered. The RBL assay solely shows a degranulation upon FceRI-cross-linking by the use of (artificial) IgE antibodies indicating their ability to receptor- and antigen-binding (meaning functionality). Reply: The reviewer seems to have especially a problem with the word ‘allergenicity’ which in his/her opinion can only be assessed with serum of allergic patients. ‘Allergenicity’ has now been replaced by ‘degranulation of mast cells’ throughout the manuscript.

As the FceRI-receptor represents the high affinity receptor for IgE it is most unlikely that IgE “are not able to bind”. Additionally, the here used “RBL-hεIa-2B12 cells” have been studied by Ladics et al. (cited in the mentioned Knipping K. et al. Intra- and inter-laboratory validation of an innovative huFcepsilonRIalpha-RBL-2H3 degranulation assay for in vitro allergenicity assessment of whey hydrolysates. Toxicol in vitro 2016) where it has been found that 80-100% of the studied RBL cell lines bound human IgE. Reply: Maybe not adequately explained in our previous reply, these humanized RBL cells (containing human IgE receptor) bind very well IgE from sera of e.g. peanut, soy, hen’s egg allergic patients. We have noticed that serum of CMA infants, usually of very young age, behave different in both the basophil activation (BAT) assay as well as the RBL-assay with less ability to bind (data not published). For safety assessment purposes, when changes are made to e.g. a CM hydrolysate, there is an urgent need of a reliable assay appropriate for screening. This cannot be achieved with serum of CMA infants (the target group, CMA adult have persistent CM allergy which is different from the transient CM allergy in infants). Therefore an epitope mapping of BLG was done with serum of 10 individual CMA infants and these epitopes were used to develop the chimeric IgE. In an internal EAACI board meeting on predictive testing (confidential) with experts from profit and non-profit organizations, this assay was assessed as a very suitable tool for assessing safety of BLG containing products. Pool sera from CMA patients are not necessarily mandatory but sera from CMA patients that are well diagnosed (by e.g. component resolved diagnostics) are available. Reply: To be sure to cover all epitopes of (in this case) BLG, a pool of CMA infants is necessary. Otherwise you might miss a change in a specific BLG epitope, which is not recognized by IgE from a single patient. Commercially available CMA serum is of the adult age category as for reasons described above not suitable for the safety assessment for CMA infants. As an extra remark: In our clinical studies with allergic infants we have performed component resolved diagnostic (ISAC) on all the subjects, also here is seen that the reactivity of the CM IgE is different (lower) than of IgE to all other allergens, as was confirmed with ImmunoCAP (not published yet).

The authors doubt the sensitivity of the here described RBL assay using sera with “1 specific allergen” but being “very specific and sensitive assay” using the chimeric antibodies. What exactly is the outcome of this assay proving “allergenicity” with chimeric (assuming mouse variable domains fused to human IgE-Fc) antibodies instead fully human IgE from patients? Why then are the developed “chimeric IgE antibodies for the use in the RBL assay for the safety assessment of whey hydrolates for CMA patients” when no data with IgE from CMA patients exist (not even in western blot analyses)? Reply: Maybe not explained adequately in the previous reply, the authors doubt the sensitivity of the use of CMA serum in this assay, not the chimeric IgE’s. As the serum of CMA allergic patients usually contains IgE to all CM proteins (all 4 caseins, BLG and ALA) which will all bind to the IgE receptor. In our case, when assessing one specific protein (BLG), the use of the chimeric IgE’s covering the complete allergic epitopes makes the assay much more sensitive than when receptors are covered with IgE against all different CM proteins. Therefore is this assay more suitable to detect small changes in a protein that could either make the protein more/less allergenic.

Moreover, as stated in lines 352 - 354 (“It is widely reported that lactosylation of proteins can reduce the allergenicity of these proteins [2-4, 6, 7, 11, 12, 14, 16, 17, 26, 27] but there is little knowledge on how lactosylation might have an effect on the allergenicity of proteins.” The question of “how”, actually, has not been answered. Reply: Agreed, it is now changed to ‘but there is little knowledge on which specific lactosylation sites might have an effect on the allergenicity of proteins.’

Minor concerns

Line 185: In a sandwich ELISA “Detection” is usually performed by the use of a secondary antibody conjugated to a e.g. chromophore. Indeed, ‘for the detection’ has been deleted, it is now: Lactosylated BLGWT was weighed, dissolved and diluted to a final concentration of 80 μg/mL in extraction buffer which is supplied in the kit.

Line 248: Please change “non-lactolated” to “non-lactosylated”. done

Line 276: Please change “percentual” to “percental”. done

Line 278: t = 0 is still not shown (“As shown in figure 5”) and Lines 276 - 278: Please explain why “the apparent most accessible lactosylation sites could be identified after 3 h incubation if compared with the non-incubated BLGWT (t = 0)”. Added to the text: In line with the intact BLG analyses, some peptides at t = 0 were identified as lactosylated, though at very low levels. These lactosylated peptides at t = 0 were set as baseline. Relative increase of lactosylation during each interval was based on the level of lactosylation at the starting point of that interval. In that way, as shown in figure 5 (and table 5, SI), the apparent most accessible lactosylation sites could be identified after 3 h incubation if compared with the non-incubated BLGWT (t = 0) (not shown). After 8 h incubation, in respect to t = 3, the already existed lactosylated peptides were observed more abundantly, plus some more lactosylation sites could be identified.

Line 298: Is it “+SD”? Please check. Yes, standard deviation

3.5 has not been revised: Assuming that the different points in time (e.g. t=0) shown in figure 7 indicate the incubation time of the BLGs with lactose it is not a “time-dependent reduction of degranulation” but rather a lactosylation-dependent reduction. Please revise point 3.5. This one slipped previous correction, ‘time-dependent’ has been deleted

Figures 6 and 7: Please specify the asterisks. Done: * One-way ANOVA p=<0.0001

Line 356: Please erase “Moreover”. done

Lines 359/360: Does lactosylation show good solubility? Please check and revise the sentence. ‘Good solubility’ was deleted. Added was: Over the whole course of the preparation of the conjugates, dimers and trimers of BLG were formed and observed on WB. Protein aggregation might have an effect on solubility as well as on antibody binding. In the case of BLG, solubility was not an issue at all time intervals, all samples were well soluble. However, reduced epitope availability as a result of aggregation is taken into account.

Lines 362 - 365: Please check syntax. done

Lines 369 - 371: Please check syntax. done

Line 370: Please specify the samples that are meant. Done, lactosylated BLGWT

Line 374: “αBLG IgE pool” is mentioned for the first time. Please clarify this “αBLG IgE pool” or paste this expression to line 115 after “pool of BLG-specific chimeric IgE monoclonal antibodies (Polpharma Biologics)”. ‘αBLG IgE pool’ replaced by ‘pool of BLG-specific chimeric IgE monoclonal antibodies’

Round 2

Reviewer 2 Report

Minor and major concerns have been revised by the authors. The explanations (see comments below) given by the authors were convincing but, unfortunately, are not included in the manuscript supposedly because they are not published yet.

Comments

The reviewer seems to have especially a problem with the word ‘allergenicity’ which in his/her opinion can only be assessed with serum of allergic patients. ‘Allergenicity’ has now been replaced by ‘degranulation of mast cells’ throughout the manuscript.

Comment: The point is that according to the European Food Safety Authority (efsa) allergenicity is “the ability to trigger an abnormal immune response that leads to an allergic reaction in a person.” and an immune response by definition means more than a degranulating cell but the involvement of additional immune/effector cells.

Maybe not adequately explained in our previous reply, these humanized RBL cells (containing human IgE receptor) bind very well IgE from sera of e.g. peanut, soy, hen’s egg allergic patients. We have noticed that serum of CMA infants, usually of very young age, behave different in both the basophil activation (BAT) assay as well as the RBL-assay with less ability to bind (data not published). For safety assessment purposes, when changes are made to e.g. a CM hydrolysate, there is an urgent need of a reliable assay appropriate for screening. This cannot be achieved with serum of CMA infants (the target group, CMA adult have persistent CM allergy which is different from the transient CM allergy in infants). Therefore an epitope mapping of BLG was done with serum of 10 individual CMA infants and these epitopes were used to develop the chimeric IgE. In an internal EAACI board meeting on predictive testing (confidential) with experts from profit and non-profit organizations, this assay was assessed as a very suitable tool for assessing safety of BLG containing products.

To be sure to cover all epitopes of (in this case) BLG, a pool of CMA infants is necessary. Otherwise you might miss a change in a specific BLG epitope, which is not recognized by IgE from a single patient. Commercially available CMA serum is of the adult age category as for reasons described above not suitable for the safety assessment for CMA infants. As an extra remark: In our clinical studies with allergic infants we have performed component resolved diagnostic (ISAC) on all the subjects, also here is seen that the reactivity of the CM IgE is different (lower) than of IgE to all other allergens, as was confirmed with ImmunoCAP (not published yet).

Comment: These are comprehensible explanations. Regrettably, in the whole manuscript this background information is missing, assumedly, because it has not been published yet. Additionally, CMA infants and CMA infants as “target group” have not been mentioned anywhere. This information would have supported the importance regarding the application of the studies described in the manuscript.  

  This manuscript is a resubmission of an earlier submission. The following is a list of the peer review reports and author responses from that submission.

Round 1

Reviewer 1 Report

In view of the rising incidence of cow's milk allergy, this manuscript is relevant and informative.

Specific comment:

  1. Title - This study reports inverse correlations between increased lactose content of BLG from cow's milk and antigenicity as well as allergenicity. Please revise the title for clarity, specifically highlighting lactolation because different carbohydrate moieties may render different effects.
  2. Design and methods - control BLG modification with sham would have been desirable to obtain the time course effect of heating at 65 ° C. Alternatively, another carbohydrate with opposite effect or no effect could have been more convincing for comparison.
  3. Results, Figure 7 legend, line 305 - please insert "surrogate marker of"  before "IgE binding ability or allergenicity"
  4. Results, Figure legend, line 306 - please state the raw value of the maximum degranulation measured by beta-hexosaminidase because the data are presented in % degranulation of maximum IgE receptor cross linking effect.

Author Response

Reviewer #1:

In view of the rising incidence of cow's milk allergy, this manuscript is relevant and informative. We thank reviewer 1 for his statement that the message of the manuscript is relevant.

Specific comment:

Title - This study reports inverse correlations between increased lactose content of BLG from cow's milk and antigenicity as well as allergenicity. Please revise the title for clarity, specifically highlighting lactolation because different carbohydrate moieties may render different effects. The title has been changed to: Limited lactosylation of beta-lactoglobulin from cow’s milk exerts strong influence on antigenicity and allergenicity

Design and methods - control BLG modification with sham would have been desirable to obtain the time course effect of heating at 65 °C. Alternatively, another carbohydrate with opposite effect or no effect could have been more convincing for comparison. On hindsight we do regret not to take along a BLG samples without sugar with the same heating process, so we do agree with the reviewer this would have been a good addition. However, as we have described in the introduction from line 69-72: beta-Lactoglobulin (beta-LG) is one of the cow's major milk proteins and the most abundant whey protein. This globular protein of about 18 kDa is folded, forming a beta-barrel (or calyx) structure. This structure is stabilized by two disulfide bonds and can be altered by heating above 65 degrees C (Taheri-Kafrani, A. et al. Effects of heating and glycation of beta-lactoglobulin on its recognition by IgE of sera from cow milk allergy patients. J Agric Food Chem 2009). Therefore we can assume that the observed effect cannot be ascribed to (thermal) denaturation as this process does not occur at our used conditions.

Results, Figure 7 legend, line 305 - please insert "surrogate marker of"  before "IgE binding ability or allergenicity" Adjusted

Results, Figure legend, line 306 - please state the raw value of the maximum degranulation measured by beta-hexosaminidase because the data are presented in % degranulation of maximum IgE receptor cross linking effect. According to a comment of another reviewer all figure legends have been elaborated: Human FceRI a chain expressing RBL cells were incubated with a pool of chimeric huIgE anti-BLG antibodies, and subsequently cross-linked with anti-hIgE antibodies which is set as maximum (100%) degranulation (bar not shown) or glycated BLG. Degranulation (surrogate marker of IgE binding ability or allergenicity) measured by release of beta -hexosaminidase (fluorescence) of glycated BLG was calculated in % of the maximum degranulation. ANOVA p=<0.0001

Reviewer 2 Report

Review of the manuscript entitled

Limited glycation of beta-lactoglobulin from cow’s milk exerts strong influence on antigenicity and allergenicity

Authors

Gerlof P. Bosman , Sergio Oliveira , Peter J. Simons , Javier Sastre Torano , Govert Somsen , Leon M. J. Knippels , Rob Haselberg , Roland J. Pieters , Johan Garssen , Karen Knipping

The manuscript entitled “Limited glycation of beta-lactoglobulin from cow’s milk exerts strong influence on antigenicity and allergenicity” by Bosman et al. describes the generation of lactosylated bovine b-lactoglobulins (BLGs) by the use of varying parameters. The modified BLGs were then analyzed by capillary electrophoresis mass spectrometry (CE-MS) to identify the lactosylated sites on the protein, and by ELISA and RBL assay to gain information for the assessment of the antigenicity as well as allergenicity of the generated lactosylated BLGs. It is concluded that an increased glycation of BLGs decreases the antigenicity as well as the allergenicity and that the reduction of the allergenicity of cow´s milk allergens can be a promising approach.

Allergy to cow´s milk (CM) is one of the most common food allergies predominantly in children. Research on one of the major allergens derived from cow´s milk (the b-lactoglobulins, BLGs) has some relevance for the nutrition of allergic infants and young children. An improved analysis of IgE-reactivities on differentially glycated BLGs might have an impact on food processing and, therefore, nutritional benefits.

The experimental set-up is robust regarding the MS-analysis and the generation of lactosylated BLGs as lactose represents almost the total potion of carbohydrates in milk.

Nevertheless, although the RBL assay provides some information on the activation of humanized rat basophilic cells, the RBL assay and the ELISA both lack data obtained with IgE from CM allergic and non-allergic individuals. This weakens a statement regarding the allergenicity of glycated BLGs, and it might be more correct to state that the analyzed BLGs provide different IgE-reactivities. On the other hand some experiments were already published by Taheri-Kafrani et al. (cited in the manuscript, [2]). Taheri-Kafrani et al. concluded that glycation has an effect on the recognition of BLG-epitopes.

Although, a number of articles regarding the elucidation of epitopes on BLGs exists, data obtained by Bosman et al. with capillary electrophoresis mass spectrometry (CE-MS) for the identification of lactosylated sites on BLG in a time-course gains deeper insights into glycation processes. This manuscript represents rather the epitope analyses of BLGs while lactosylation processes than allergic aspects.

A weak point is that the performed RBL assay using chimeric antibodies is not sufficient to show allergenicity. The authors should emphasize a possible benefit of their results regarding nutrition in the context of allergic reactions to CM.

Major concerns

  • General: Figure legends are not sufficient; please describe the figures in detail (what information is depicted by the figures).
  • General: Figures need to be revised. Please add axis labels, how often have experiments been conducted. For time points add hours, for example
  • General: Controls regarding heat-treated BLGs without lactose are missing.

To make sure that the decreased recognition of BLGs by antibodies is due to masked epitope by lactosylation controls with BLGs receiving the same treatment but without added lactose are missing. These controls should exclude a decreased recognition due to epitopes compromised by the heat treatment.

  • Lines 284/285 and figure 6: Assuming that the different points in time (e.g. t=0) shown in figure 6 indicate the incubation time of the BLGs with lactose it is not a “time-dependent reduction of antigenicity” but rather a lactosylation-dependent reduction. Please revise point 3.4.
  • Figure 6, lines 291/292 and 3.4: Please elucidate the shown “significant decrease”. E.g. what statistical test was performed?
  • Lines 296/297: The ELISA does not show a difference of glycation between t = 16 and WB because WB is not shown. Please elaborate the ELISA performance and results and adapt it to the results of the RBL assay. Please explain WB (this abbreviation is most commonly used for western blot, see line 141).
  • General: The results show a reduced IgE-binding capacity. Allergenicity should be determined by the use of serum from allergic and non-allergic patients (see also line 305).
  • Lines 356/357: The ELISA does not show a time course. Please revise.

Minor concerns

  • General: It should read “lactosylation”, not “lactolation”, due to the lactosyl group of lactose.
  • General: space between number and °C.
  • Line 8: Please change “treatments” to “treatment”.
  • Lines 20/21: Primary objective starts without “To” but lowercased “analyze”; Secondary objective starts with “To”, please adjust.
  • Lines 26/27: Redundant “human Fce receptor type 1 FceRI complex”; please change to “human Fce receptor type 1 (FceRIa)”.
  • Line 43: „carbonyl“; please change to „carbonyl compound“.
  • Lines 132 ff.: Lactolation of BLGA and BLGB is not described; please describe sample preparation for BLGA and BLGB in detail, comparable to BLGWT.
  • Line 144: “membrane. PVDF”; one additional space to remove.
  • Lines 166-171: results are described in the methods section; please describe the methods applied.
  • Line 171: “18277.07 Da respectively.”; please add a comma between “Da” and “ respectively”.
  • Line 123 and lane 163: Please describe the sample preparation for “2.5 Intact protein analysis” Was the sample preparation conducted without DTT as it is assumed that all pairs of cysteines form cystines?
  • Lines 185 and 188: Please specify the proteins that were diluted.
  • Lines 185, 282 and 284: Please comment on “glycated BLG” and “Glycation”. Is lactosylated and lactosylation meant?
  • Lines 185, 282 and 284: Please explain the specificity of the used ELISA.
  • Lines 191/192: see lines 26/27
  • Line 198: “RBL cells”: please clarify if these RBL cells are the same as the previously mentioned “RBL-huFcεRI cells”
  • Line 204: Please explain statistics more detailed (see also comments on figure 6).
  • Figure 1: What type of BLG is shown (BLGA, BLGB or BLGWT)?
  • Figure 1: A protein molecular weight marker is missing.
  • Figure 1: Please specify “Reference BLG” (bought from a company?).
  • Figure 1: The arrow indicating the “BLG trimer ≈ 55,2 kDa” indicates a positive band at a similar height of a glycated BLG after 24 h. Please check and revise.
  • Figure 1: Please use decimal points (e.g. 55.2 kDa).
  • Line 225: Please change “minutes” to “min”. State abbreviations when mentioned for the first time in the manuscript.
  • Line 226: Please mention “background electrolyte (BGE)” (line 231) here.
  • Line 229: Please write the full word for “EOF”.
  • Lines 232: Please explain “(R=0.88)”.
  • Figure 2, line 239 and table 3 (supplementary): The masses do not match: Please check.
  • SI figure 1: “(Bottom) Deconvoluted spectrum of BLGA.” This figure is already shown in the manuscript.
  • SI figure 1: It is stated that BLGA and BLGB separation was observed. Please explain which part of the figure indicates that observation.
  • Table 3 of the SI: The represented sequences are neither included in table 3 nor specified. Please describe the sequences separately and go into detail regarding the underlined letters. Which program was used to calculate the theoretical masses?
  • Line 242-261: Figures 3 and 4 are not mentioned in the text. Please indicate.
  • Line 256: According to “15 lysine and 3 arginine” please specify the meant amino acid of the N-terminus.
  • Table 5, SI: Please change “Percentual” to “Percental”.
  • Table 5, SI: Please explain the green highlighted numbers.
  • Lines 271-275: Which BLG is shown?
  • Lines 271-275: t = 0 h is not shown.
  • Lines 271-275: t = 8 h shows the most accessible lactolation sites. Please explain why it is t = 3 h.
  • Lines 285 and 286: Please insert spaces between “t”, “=” and the corresponding number.
  • Figure 6: Please specify the here used BLG (BLGA? BLGB? BLGWT?).
  • Figure 6: Please write the full word for “gBLG” the first time of appearance
  • Line 293: Please change “3.6” to “3.5”.
  • 6 Allergenicity of glycated BLG using RBL-hεIa-2B12 cells: Please revise this part according to “3.4. Antigenicity of glycated BLG using ELISA”
  • Line 315: Please use the plural form of “MR”.
  • Line 319: Please use the plural form of “epitope”.
  • Line 331: Please specify the amino acids of the mentioned “N-terminal”.
  • Lines 350-352: This sentence is redundant (see previous sentence).
  • Line 354: “solubility” is mentioned in the manuscript for the first time. Please check and revise.
  • Line 355: Please explain “in time”.
  • Lines 353-357: Please revise the sentence and check syntax.
  • Line 361: Please write “RBL-cells it was”.
  • Line 361: Please specify the molecule that is meant.
  • Line 363: Please change “The” to “the”.
  • Line 363: Please change “SDS-PAGE” to “western blot”.
  • Line 364: In line 129 it is stated that the samples for 0 h were heated. Please check and revise.
  • Line 366: Please adjust “ɑBLG IgE”.

Author Response

Limited glycation of beta-lactoglobulin from cow’s milk exerts strong influence on antigenicity and allergenicity. Authors: Gerlof P. Bosman , Sergio Oliveira , Peter J. Simons , Javier Sastre Torano , Govert Somsen , Leon M. J. Knippels , Rob Haselberg , Roland J. Pieters , Johan Garssen , Karen Knipping

The manuscript entitled “Limited glycation of beta-lactoglobulin from cow’s milk exerts strong influence on antigenicity and allergenicity” by Bosman et al. describes the generation of lactosylated bovine b-lactoglobulins (BLGs) by the use of varying parameters. The modified BLGs were then analyzed by capillary electrophoresis mass spectrometry (CE-MS) to identify the lactosylated sites on the protein, and by ELISA and RBL assay to gain information for the assessment of the antigenicity as well as allergenicity of the generated lactosylated BLGs. It is concluded that an increased glycation of BLGs decreases the antigenicity as well as the allergenicity and that the reduction of the allergenicity of cow´s milk allergens can be a promising approach.

Allergy to cow´s milk (CM) is one of the most common food allergies predominantly in children. Research on one of the major allergens derived from cow´s milk (the b-lactoglobulins, BLGs) has some relevance for the nutrition of allergic infants and young children. An improved analysis of IgE-reactivities on differentially glycated BLGs might have an impact on food processing and, therefore, nutritional benefits. The experimental set-up is robust regarding the MS-analysis and the generation of lactosylated BLGs as lactose represents almost the total potion of carbohydrates in milk. Nevertheless, although the RBL assay provides some information on the activation of humanized rat basophilic cells, the RBL assay and the ELISA both lack data obtained with IgE from CM allergic and non-allergic individuals. This weakens a statement regarding the allergenicity of glycated BLGs, and it might be more correct to state that the analyzed BLGs provide different IgE-reactivities. The authors do not agree with this statement of the reviewer. In the past we have attempted to standardize the RBL-assay with serum of cow’s milk allergic (CMA) patients, but have not been very successful. This is due to 1) most CMA patients are young infants, IgE titers are usually low and the amount of blood you can withdraw are very small. 2) Even if CM IgE titers are high, for unknown reasons they not always are able to bind the IgE receptors. 3) Individual sera might have IgE against 1 of a few allergenic epitopes, but does not cover all allergenic epitopes in a protein, therefore you need to use a pool of CMA sera, knowing to cover all epitopes. 4) CMA patients usually have IgE against different cow’s milk proteins, and not only 1 of the proteins. Using sera with IgE against e.g. casein, beta-lactoglobulin and alpha-lactalbumin make the assay less sensitive when investigating 1 specific allergen. For all the above reasons we have developed the chimeric IgE antibodies for the use in the RBL-assay for the safety assessment of whey hydrolysates for CMA patients. We know the pool of 6 chimeric antibodies cover the complete allergenic epitopes of BLG, as investigated by epitope mapping against serum of CMA patients (Knipping K. et al. Development of beta-lactoglobulin-specific chimeric human IgEkappa monoclonal antibodies for in vitro safety assessment of whey hydrolysates. PLoS One 2014). The assay was then validated in a ringtrial using (blinded) whey hydrolysates with known (pre)clinical outcome and showed a very good predictability of allergenicity (Knipping K. et al. Intra- and inter-laboratory validation of an innovative huFcepsilonRIalpha-RBL-2H3 degranulation assay for in vitro allergenicity assessment of whey hydrolysates. Toxicol in vitro 2016) and is now used by several manufacturers of hydrolysates for the ultimate safety assessment. Therefore we believe this RBL-assay using the chimeric antibodies is a very specific and sensitive assay to assess allergenicity of BLG.

On the other hand some experiments were already published by Taheri-Kafrani et al. (cited in the manuscript, [2]). Taheri-Kafrani et al. concluded that glycation has an effect on the recognition of BLG-epitopes. Although, a number of articles regarding the elucidation of epitopes on BLGs exists, data obtained by Bosman et al. with capillary electrophoresis mass spectrometry (CE-MS) for the identification of lactosylated sites on BLG in a time-course gains deeper insights into glycation processes. This manuscript represents rather the epitope analyses of BLGs while lactosylation processes than allergic aspects. A weak point is that the performed RBL assay using chimeric antibodies is not sufficient to show allergenicity. The authors should emphasize a possible benefit of their results regarding nutrition in the context of allergic reactions to CM. We thank reviewer 2 for the very valuable contribution and we believe that this has greatly increased the quality of our manuscript.

Major concerns

General: Figure legends are not sufficient; please describe the figures in detail (what information is depicted by the figures). Figure legends have been revised and details have been added.

General: Figures need to be revised. Please add axis labels, how often have experiments been conducted. For time points add hours, for example Figures have been revised, details have been added. The n of experiments have been added to the figure legend and not the figure itself.

General: Controls regarding heat-treated BLGs without lactose are missing. To make sure that the decreased recognition of BLGs by antibodies is due to masked epitope by lactosylation controls with BLGs receiving the same treatment but without added lactose are missing. These controls should exclude a decreased recognition due to epitopes compromised by the heat treatment. On hindsight we do regret not to take along a BLG samples without sugar with the same heating process, so we do agree with the reviewer this would have been a good addition. However, as we have described in the introduction from line 69-72: beta-Lactoglobulin (beta-LG) is one of the cow's major milk proteins and the most abundant whey protein. This globular protein of about 18 kDa is folded, forming a beta-barrel (or calyx) structure. This structure is stabilized by two disulfide bonds and can be altered by heating above 65 degrees C (Taheri-Kafrani, A. et al. Effects of heating and glycation of beta-lactoglobulin on its recognition by IgE of sera from cow milk allergy patients. J Agric Food Chem 2009). Therefore we can assume that the observed effect cannot be ascribed to (thermal) denaturation as this process does not occur at our used conditions.

Lines 284/285 and figure 6: Assuming that the different points in time (e.g. t=0) shown in figure 6 indicate the incubation time of the BLGs with lactose it is not a “time-dependent reduction of antigenicity” but rather a lactosylation-dependent reduction. Please revise point 3.4. ‘Time-dependent’ has been changed to ‘lactosylation-dependent’.

Figure 6, lines 291/292 and 3.4: Please elucidate the shown “significant decrease”. E.g. what statistical test was performed? Changed to ‘Lactosylation showed a significant (ANOVA p=0.0001) lactosylation-dependent reduction of antigenicity (figure 6).

Lines 296/297: The ELISA does not show a difference of glycation between t = 16 and WB because WB is not shown. Please elaborate the ELISA performance and results and adapt it to the results of the RBL assay. Please explain WB (this abbreviation is most commonly used for western blot, see line 141). We think the reviewer means ‘RBL’ and not ‘WB’ since t = 16 h in WB is shown. In a first step, all samples were tested on the WB and BLG ELISA and in both assays was seen that there was no difference in pattern (WB) and binding to IgG (BLG ELISA) between 16 and 24 h. Since there was a limited amount of chimeric IgE antibodies available, it was decided to leave out the 24 h sample.

General: The results show a reduced IgE-binding capacity. Allergenicity should be determined by the use of serum from allergic and non-allergic patients (see also line 305).’ The authors do not agree with this statement of the reviewer. In the past we have attempted to standardize the RBL-assay with serum of cow’s milk allergic (CMA) patients, but have not been very successful. This is due to 1) most CMA patients are young infants, IgE titers are usually low and the amount of blood you can withdraw are very small. 2) Even if CM IgE titers are high, for unknown reasons they not always are able to bind the IgE receptors. 3) Individual sera might have IgE against 1 of a few allergenic epitopes, but does not cover all allergenic epitopes in a protein, therefore you need to use a pool of CMA sera, knowing to cover all epitopes. 4) CMA patients usually have IgE against different cow’s milk proteins, and not only 1 of the proteins. Using sera with IgE against e.g. casein, beta-lactoglobulin and alpha-lactalbumin make the assay less sensitive when investigating 1 specific allergen. For all the above reasons we have developed the chimeric IgE antibodies for the use in the RBL-assay for the safety assessment of whey hydrolysates for CMA patients. We know the pool of 6 chimeric antibodies cover the complete allergenic epitopes of BLG, as investigated by epitope mapping against serum of CMA patients (Knipping K. et al. Development of beta-lactoglobulin-specific chimeric human IgEkappa monoclonal antibodies for in vitro safety assessment of whey hydrolysates. PLoS One 2014). The assay was then validated in a ringtrial using (blinded) whey hydrolysates with known (pre)clinical outcome and showed a very good predictability of allergenicity (Knipping K. et al. Intra- and inter-laboratory validation of an innovative huFcepsilonRIalpha-RBL-2H3 degranulation assay for in vitro allergenicity assessment of whey hydrolysates. Toxicol in vitro 2016) and is now used by several manufacturers of hydrolysates for the ultimate safety assessment. Therefore we believe this RBL-assay using the chimeric antibodies is a very specific and sensitive assay to assess allergenicity of BLG.

Lines 356/357: The ELISA does not show a time course. Please revise. Changed to ‘For the antigenicity assessment using the BLG ELISA we found that lactosylation with a temperature of 65 °C showed good solubility and a significant reduction of IgG antibody-binding capacity with increased lactosylation,’

Minor concerns All minor concerns have been addressed and adjusted accordingly, if necessary a comment was added

General: It should read “lactosylation”, not “lactolation”, due to the lactosyl group of lactose.

General: space between number and °C.

Line 8: Please change “treatments” to “treatment”.

Lines 20/21: Primary objective starts without “To” but lowercased “analyze”; Secondary objective starts with “To”, please adjust.

Lines 26/27: Redundant “human Fce receptor type 1 FceRI complex”; please change to “human Fce receptor type 1 (FceRIa)”.

Line 43: „carbonyl“; please change to „carbonyl compound“.

Lines 132: Lactolation of BLGA and BLGB is not described; please describe sample preparation for BLGA and BLGB in detail, comparable to BLGWT.

Line 144: “membrane. PVDF”; one additional space to remove.

Lines 166-171: results are described in the methods section; please describe the methods applied.

Line 171: “18277.07 Da respectively.”; please add a comma between “Da” and “ respectively”.

Line 123 and lane 163: Please describe the sample preparation for “2.5 Intact protein analysis” Was the sample preparation conducted without DTT as it is assumed that all pairs of cysteines form cystines?

Lines 185 and 188: Please specify the proteins that were diluted.

Lines 185, 282 and 284: Please comment on “glycated BLG” and “Glycation”. Is lactosylated and lactosylation meant?

Lines 185, 282 and 284: Please explain the specificity of the used ELISA. The supplier of the BLE ELISA from ELISA systems let us know: The antibody we use is a polyclonal antibody raised against BLG purified from bovine milk (>90% pure by SDS-PAGE). This is added to the M&M 2.7. Antigenicity assessment of BLG line 187.

Lines 191/192: see lines 26/27

Line 198: “RBL cells”: please clarify if these RBL cells are the same as the previously mentioned “RBL-huFcεRI cells”

Line 204: Please explain statistics more detailed (see also comments on figure 6). We could only perform statistical analysis on the data from the BLG ELISA and the RBL-assay. These were done with a one-way ANOVA which is used to determine whether there are any statistically significant differences between the means of two or more independent groups, which is stronger than a student’s t-test. We have added ‘one-way’ to ANOVA and explained in more detail why ANOVA was used and for which data we performed statistical analysis.

Figure 1: What type of BLG is shown (BLGA, BLGB or BLGWT)? BLGWT, is added to the legend of figure 1

Figure 1: A protein molecular weight marker is missing. A prestained MW was used and is added to the figure and M&M

Figure 1: Please specify “Reference BLG” (bought from a company?).

Figure 1: The arrow indicating the “BLG trimer ≈ 55,2 kDa” indicates a positive band at a similar height of a glycated BLG after 24 h. Please check and revise. We noticed that during running of the gel, more lactosylation had a slight impact on the electrophoreses. Although in the reference samples the band above the BLG trimer seems to be at the same height as the lactosylated trimer of the 24 h sample, this is not the case. When comparing to the MW markers and BLG t = 0 h, this band is also visible but less pronounced. Also the lactosylated sample show this extra band. Not sure whether this would be the explanation, but this band seems to be at the height of just below 75 kDa marker, and 4x18,4 = 73,6 so it might be 4xBLGs.

Figure 1: Please use decimal points (e.g. 55.2 kDa).

Line 225: Please change “minutes” to “min”. State abbreviations when mentioned for the first time in the manuscript.

Line 226: Please mention “background electrolyte (BGE)” (line 231) here.

Line 229: Please write the full word for “EOF”.

Lines 232: Please explain “(R=0.88)”.

Figure 2, line 239 and table 3 (supplementary): The masses do not match: Please check.

SI figure 1: “(Bottom) Deconvoluted spectrum of BLGA.” This figure is already shown in the manuscript.

SI figure 1: It is stated that BLGA and BLGB separation was observed. Please explain which part of the figure indicates that observation.

Table 3 of the SI: The represented sequences are neither included in table 3 nor specified. Please describe the sequences separately and go into detail regarding the underlined letters. Which program was used to calculate the theoretical masses?

Line 242-261: Figures 3 and 4 are not mentioned in the text. Please indicate.

Line 256: According to “15 lysine and 3 arginine” please specify the meant amino acid of the N-terminus.

Table 5, SI: Please change “Percentual” to “Percental”.

Table 5, SI: Please explain the green highlighted numbers.

Lines 271-275: Which BLG is shown?

Lines 271-275: t = 0 h is not shown.

Lines 271-275: t = 8 h shows the most accessible lactolation sites. Please explain why it is t = 3 h.

Lines 285 and 286: Please insert spaces between “t”, “=” and the corresponding number.

Figure 6: Please specify the here used BLG (BLGA? BLGB? BLGWT?).

Figure 6: Please write the full word for “gBLG” the first time of appearance

Line 293: Please change “3.6” to “3.5”.

Allergenicity of glycated BLG using RBL-hεIa-2B12 cells: Please revise this part according to “3.4. Antigenicity of glycated BLG using ELISA”

Line 315: Please use the plural form of “MR”.

Line 319: Please use the plural form of “epitope”.

Line 331: Please specify the amino acids of the mentioned “N-terminal”.

Lines 350-352: This sentence is redundant (see previous sentence).

Line 354: “solubility” is mentioned in the manuscript for the first time. Please check and revise.

Line 355: Please explain “in time”.

Lines 353-357: Please revise the sentence and check syntax.

Line 361: Please write “RBL-cells it was”.

Line 361: Please specify the molecule that is meant.

Line 363: Please change “The” to “the”.

Line 363: Please change “SDS-PAGE” to “western blot”.

Line 364: In line 129 it is stated that the samples for 0 h were heated. Please check and revise.

Line 366: Please adjust “ɑBLG IgE”

Reviewer 3 Report

This manuscript contains the results of experiments conducted to evaluate the effect of lactosylation on the antigenicity and allergenicity of β-lactoglobulin, one of the major milk allergens.  Lactosylation is a phenomenon that occurs when milk is heat-processed resulting from the Maillard reaction occurring between lactose (a reducing sugar) and amino moieties present in the structure of BLG.  This manuscript is very well written, the experimental approaches are well presented, and the results are thoroughly explained.  Because the study is well conducted and well presented, this reviewer has only two general comment and one specific correction.

General comments:

  • On lines 53-54, the authors mention that they are interested in insights into the effect of glycation on the development of food allergies. However, the experiments conducted the BLG evaluated the effect of glycation on the elicitation (mediator release) phase of the allergic response.  These effects are interesting and worthy of publication.  However, the authors should note somewhere that they are NOT evaluating the effect of glycation on  the sensitizing potential of BLG.  A comparison of the sensitizing potential of glycated BLG vs. native BLG in an animal model would be a rather interesting follow-up investigation.
  • The authors evaluated the comparative antigenicity of glycated BLG using a commercial ELISA kit for BLG that has IgG antisera specific to BLG (presumably native BLG but that might depend upon how the kit manufacturers raised the antisera – the nature of the antigen – was it heat-processed? ELISA kits rely upon solution chemistry and decreased IgG binding can occur because of either chemical or physical modification of the protein of interest or because the protein has become insoluble (through aggregation usually) and is no longer in solution.  On line 354, the authors mention that they encountered good solubility in the ELISA experiments.  They should consider expanding the discussion of solubility of glycated vs. native BLG.

Specific comments:

  • Line 18:  treatments should be singular.

Author Response

This manuscript contains the results of experiments conducted to evaluate the effect of lactosylation on the antigenicity and allergenicity of β-lactoglobulin, one of the major milk allergens.  Lactosylation is a phenomenon that occurs when milk is heat-processed resulting from the Maillard reaction occurring between lactose (a reducing sugar) and amino moieties present in the structure of BLG.  This manuscript is very well written, the experimental approaches are well presented, and the results are thoroughly explained. Because the study is well conducted and well presented, this reviewer has only two general comment and one specific correction. We thank reviewer 3 for his statement that the message of the manuscript is relevant.

General comments:

On lines 53-54, the authors mention that they are interested in insights into the effect of glycation on the development of food allergies. However, the experiments conducted the BLG evaluated the effect of glycation on the elicitation (mediator release) phase of the allergic response. These effects are interesting and worthy of publication. However, the authors should note somewhere that they are NOT evaluating the effect of glycation on the sensitizing potential of BLG. A comparison of the sensitizing potential of glycated BLG vs. native BLG in an animal model would be a rather interesting follow-up investigation. Indeed the investigation of this study is solely focused on the elicitation phase and not the sensitization phase of allergy. Although we are also interested in the ability of the lactosylated BLG to prevent allergic sensitization, we decided to remove all refers to this sensitization phase to keep the manuscript focused on only the elicitation phase. Therefore line 53-54 has now changed to: Ample evidence is available about the pathogenesis caused by glycation in the context of various diseases but there are a few studies that provide a thorough insight into the impact of glycation in food allergies.

The authors evaluated the comparative antigenicity of glycated BLG using a commercial ELISA kit for BLG that has IgG antisera specific to BLG (presumably native BLG but that might depend upon how the kit manufacturers raised the antisera – the nature of the antigen – was it heat-processed? ELISA kits rely upon solution chemistry and decreased IgG binding can occur because of either chemical or physical modification of the protein of interest or because the protein has become insoluble (through aggregation usually) and is no longer in solution.  On line 354, the authors mention that they encountered good solubility in the ELISA experiments.  They should consider expanding the discussion of solubility of glycated vs. native BLG. In preparation of the samples no solubility issues have been encountered. In fact, all samples from each time interval were well soluble. If aggregation - as a result of incubation at elevated temperatures for extended time periods - would affect outcomes in ELISA, due to reduced epitope availability, similar inhibition of signals is expected as equal aggregation was observed at each of the intervals on WB. A brief explanation is added to the main text.

Specific comments:

Line 22:  treatments should be singular. Adjusted